# *PPP1R7* Is a Novel Translocation Partner of *CBFB* via t(2;16)(q37;q22) in Acute Myeloid Leukemia

**DOI:** 10.3390/genes13081367

**Published:** 2022-07-29

**Authors:** Lulu Wang, Wei Wang, Hannah C. Beird, Xueqian Cheng, Hong Fang, Guilin Tang, Gokce A. Toruner, C. Cameron Yin, M. James You, Ghayas C. Issa, Gautam Borthakur, Guang Peng, Joseph D. Khoury, L. Jeffrey Medeiros, Zhenya Tang

**Affiliations:** 1Departments of Clinical Cancer Prevention, The University of Texas MD Anderson Cancer Center, Houston, TX 77030, USA; wanglulu@tmu.edu.cn (L.W.); xcheng6@mdanderson.org (X.C.); gpeng@mdanderson.org (G.P.); 2Departments of Hematopathology, The University of Texas MD Anderson Cancer Center, Houston, TX 77030, USA; wwang13@mdanderson.org (W.W.); hfang@mdanderson.org (H.F.); gtang@mdanderson.org (G.T.); gatoruner@mdanderson.org (G.A.T.); cyin@mdanderson.org (C.C.Y.); mjamesyou@mdanderson.org (M.J.Y.); jkhoury@mdanderson.org (J.D.K.); ljmedeiros@mdanderson.org (L.J.M.); 3Departments of Genomic Medicine, The University of Texas MD Anderson Cancer Center, Houston, TX 77030, USA; hccheung@mdanderson.org; 4Departments of Leukemia, The University of Texas MD Anderson Cancer Center, Houston, TX 77030, USA; gcissa@mdanderson.org (G.C.I.); gborthak@mdanderson.org (G.B.)

**Keywords:** *CBFB* rearrangement, novel partner gene, *PPP1R7*, microhomology, AML

## Abstract

In a subset of acute myeloid leukemia (AML) cases, the core binding factor beta subunit gene (*CBFB*) was rearranged via inv(16)(p13.1q22) or t(16;16)(p13.1;q22), in which the smooth muscle myosin heavy chain 11 gene (*MYH11*) was the partner (*CBFB::MYH11*). Rare variants of *CBFB* rearrangement occurring via non-classic chromosomal aberrations have been reported, such as t(1;16), t(2;16), t(3;16), t(5;16), and t(16;19), but the partners of *CBFB* have not been characterized. We report a case of AML with a complex karyotype, including t(2;16)(q37;q22), in which the protein phosphatase 1 regulatory subunit 7 gene (*PPP1R7*) at chromosome 2q37 was rearranged with *CBFB* (*CBFB::PPP1R7*). This abnormality was inconspicuous by conventional karyotype and interphase fluorescence in situ hybridization (FISH), thus leading to an initial interpretation of inv(16)(p13.1q22); however, metaphase FISH showed that the *CBFB* rearrangement involved chromosome 2. Using whole genome and Sanger sequencing, the breakpoints were identified as being located in intron 5 of *CBFB* and intron 7 of *PPP1R7*. A microhomology of CAG was found in the break and reconnection sites of *CBFB* and *PPP1R7*, thus supporting the formation of *CBFB::PPP1R7* by microhomology-mediated end joining.

## 1. Introduction

Core binding factor beta subunit gene (*CBFB*) is located on chromosome 16q22 and encodes the beta subunit of core binding factor (CBF), a member of the PEBP2/CBF transcription factor family considered to be a master regulator of normal hematopoiesis [1,2,3,4,5]. CBF also has an alpha subunit (CBFA), which binds to DNA and is composed of three subunits, RUNX1, RUNX2, and RUNX3 [6,7]. By contrast, CBFB is a non-DNA binding unit that is required for DNA binding by CBFA. Genetic alterations of these genes (*CBFB*, *RUNX1*, *RUNX2*, and *RUNX3*) can alter normal binding to DNA by CBF, thus resulting in a failure of hematopoiesis or hematologic malignancies, particularly acute myeloid leukemia (AML) [8,9,10,11,12,13].

Chromosomal aberrations causing *CBFB* rearrangement, such as the pericentric inversion inv(16)(p13.1q22) and, less frequently, the t(16;16)(p13.1;q22), hereafter referred as inv(16)/t(16;16), lead to a rearrangement between *CBFB* and smooth muscle myosin heavy chain 11 gene (*MYH11*) located at 16p13.1, thus resulting in a fusion transcript consisting of 5′-*CBFB* and 3′-*MYH11* (*CBFB::MYH11*) [1,3,5]. The chimeric CBFB/MYH11 protein can alter CBF binding to enhancers and promotors, resulting in dysregulation of the transcription program and consequently leukemogenesis, although the latter requires additional mutational events [5,13,14,15]. Cases of AML with inv(16)/t(16;16); *CBFB::MYH11* and AML with t(8;21)(q22;q22); *RUNX1::RUNX1T1* are included in the umbrella category of CBF AML. This subset of neoplasms represents 10 to 15% of all AML cases and is considered a favorable risk, if appropriate intensive chemotherapy regimens are administered [16].

Chromosomal aberrations involving 16q22 other than classic inv(16)/t(16;16) have been reported rarely in AML cases in the literature, including t(1;16)(p32;q22) [17], t(1;16)(q43;q22) [18], t(3;16)(q21;q22) [19], t(3;16)(q24;q22) [20], t(5;16)(q13;q22) [21], and t(5;16)(q33;q22) [22,23]. Some of these translocations may imply potential *CBFB* rearrangement with a novel partner gene, other than *MYH11*. However, none of these cases has been assessed in depth to determine potential partners of *CBFB*.

Recently, we reviewed our files for cases of AML with *CBFB* rearrangement and identified three unusual AML cases with t(1;16)(q21;q22), t(2;16)(q37;q22), or t(16;19)(q22;q13.3) [24]. Fluorescence in situ hybridization (FISH) analysis in these cases showed a 3′-*CBFB* signal located on the chromosomal loci 1q21, 2q37, and 19q13.3, respectively. In all three cases, a cryptic *CBFB::MYH11* rearrangement was excluded by reverse transcriptase polymerase chain reaction (RT-PCR) methods. Therefore, a novel partner of *CBFB* rearrangement is very likely in these three AML cases. [24]

In the current study, we performed whole genome sequencing (WGS) and Sanger sequencing in one of these AML cases, associated with t(2;16)(q37;q22), which had available DNA for further investigation. We identified the protein phosphatase 1 regulatory subunit 7 gene (*PPP1R7*) located on chromosome 2q37 as a novel partner of *CBFB* via the translocation [24]. As far as we know, this report is the first to discover a novel partner for *CBFB*, other than *MYH11*.

## 2. Materials and Methods

### 2.1. Patients

We searched the database of the Clinical Cytogenetics Laboratory in the Department of Hematopathology, the University of Texas MD Anderson Cancer Center, for cases assessed by CBFB break-apart (BAP) FISH testing performed from 1 June 2000 through 31 May 2021. From a cohort of 271 AML cases with a confirmed *CBFB* rearrangement by FISH and/or RT-PCR, 3 cases were identified as negative for *CBFB::MYH11* fusion [24]. The *CBFB* rearrangement in these cases was derived from non-classic t(1;16), t(2;16), and t(16;19), respectively. Of these three cases, in one case DNA was available for additional analysis. The electronic medical records were reviewed, and clinicopathologic and other laboratory information were collected. We performed a literature search and identified 7 AML cases with potential and/or confirmed *CBFB* rearrangement via translocations other than inv(16)/t(16;16). This study was approved by the Institutional Review Board (IRB) of the University of Texas MD Anderson Cancer Center.

### 2.2. Chromosomal Analysis

Conventional G-banded chromosomal analysis (karyotyping) was performed on 24 and 48 h cell cultures of bone marrow (BM) aspirate without mitogens. Twenty metaphases were analyzed routinely for each specimen, and the final results were reported following the 2020 International System for Human Cytogenetics Nomenclature (ISCN 2020) guidelines [24,25].

### 2.3. FISH Analysis

The Vysis CBFB Dual Color Break Apart Rearrangement Probe (Abbott Molecular, Des Plaines, IL, USA) was applied for FISH analysis. To better understand the exact location of the 5′ and 3′ *CBFB* signals and correlate with findings of chromosomal analysis, mapback FISH analyses on G-banded and karyotyped metaphases were performed [24,25].

### 2.4. Whole Genome Sequencing

The QIAamp DNA mini kit (QIAGEN, Germantown, MD, USA) was used for genomic DNA (gDNA) extraction from bone marrow aspirate at diagnosis. A total of 50 ng of purified gDNA was fragmented to 600–1000 bp sizes and underwent adapter ligation, indexing, and amplification. The SureSelectQXT reagent kit (Agilent Technologies Inc., Santa Clara, CA, USA) was used for DNA library prep. Low-pass whole-genome sequencing (WGS) was performed as reported previously. Briefly, the mean target coverage was 8.86× with a mapping rate of 91.27%. Alignment to the GRCh38/hg38 reference genome was performed by using BWA13 and BWA-MEM, as reported previously [26]. The HMMcopy software (Bioconductor. DOI: 10.18129/B9.bioc.HMMcopy, (accessed on 20 June 2022)) [27] was applied to analyze the copy number status with focus on 2q, 3q, and 16q that were involved in translocations in our case.

#### PCR and Sanger Sequencing

The KOD Xtreme™ Hot Start DNA Polymerase (MilliporeSigma Life Science, Burlington, NJ, USA) was applied for PCR amplification. Approximately 30 ng gDNA was used in each PCR reaction. According to WGS analysis results, the following primer pair was designed and applied for PCR amplification: forward primer, PPP1R7_F1: 5′-CAGTCCTCAGTATGCAGGTACG-3′ (chr2:241662491–241662512) and reverse primer, CBFB_R1: 5′-GCTGAAAACTCTTCATGGGAAA-3′ (chr16:72887661–72887640) (all coordinates in this study are following the GRCh38/hg38, T2T-CHM13v2.0 assembly). The PCR products were recovered from 1.5% agarose gel and then purified with QIAquick Gel Extraction Kit (QIAGEN, Germantown, MD, USA). The purified PCR products were ligated to Zero Blunt TOPO PCR Cloning Kit (Life Technologies, Carlsbad, CA, USA); then, they were subjected to Sanger sequencing using the Sequencing Primer M13R at the core facility of our institution. The obtained DNA sequences were analyzed and aligned using DNASTAR Lasergene 17 SeqManPro (DNASTAR, Inc., Madison, WI, USA) and SnapGene Viewer (GSL Biotech LLC, San Diego, CA, USA).

## 3. Results

### 3.1. Case Report

The patient was a 58-year-old man who had a history of thyroiditis and intermittent cardiac arrhythmia. He was diagnosed initially with AML associated with inv(16) at another hospital and treated with cytarabine and mitoxantrone for induction chemotherapy, followed by two courses of high-dose cytarabine and mitoxantrone for consolidation chemotherapy. He achieved a complete remission for approximately one year. He then developed relapsed disease and was referred to our institution. The complete blood count at this time showed hemoglobin 14.5 g/dL (normal, 12.5–17.3 g/dL), hematocrit 41.9% (normal, 38.3–51.2%), platelets 95 K/uL (140–400 K/uL), and white blood cell (WBC) count 4.9 K/uL (normal, 4–11 K/uL), with 74% neutrophils, 18% lymphocytes, 7% monocytes, and 1% eosinophils. The patient underwent bone marrow (BM) examination at this time.

BM aspirate smears showed 12% blasts and 1% eosinophils. The blasts were intermediate to large size with oval-to-irregular nuclei and small-to-moderate amounts of cytoplasm. Some blasts had numerous cytoplasmic granules, and a few had vacuoles. Atypical eosinophil precursors with basophilic granules were seen but were rare (Figure 1). Blasts were partially positive for myeloperoxidase by cytochemical staining. The bone marrow biopsy specimen showed 20% cellularity, with decreased megakaryocytes and increased immature cells consistent with blasts presenting in small clusters. Flow cytometry immunophenotypic analysis showed increased blasts positive for CD13, CD33, CD34, CD38 (bright), CD117, HLA-DR, and myeloperoxidase, as well as negative for CD14, CD41, CD56, CD64, TdT, and B- and T-cell markers. These findings were diagnostic for recurrent/relapsed AML.

The patient was treated with a regimen of cytarabine plus/minus cisplatin plus/minus cloretazine, followed by a matched related-donor peripheral blood stem cell transplant (SCT) during second remission. The patient had several follow-up visits with no evidence of disease relapse. However, he died from unknown causes approximately 10 months after his discharge from our hospital.

### 3.2. Conventional Cytogenetics, FISH and Molecular Results

Conventional chromosomal analysis of the BM showed a complex karyotype: 50, XY, t(2;16)(q37;q22), t(3;16)(p21;p13), +8, +21, +22, +mar[5]/51, idem[cp2]/46, XY[13]. A representative karyogram is illustrated in Figure 2. Concurrent CBFB BAP FISH showed *CBFB* rearrangement in 40 of 200 (20%) cells analyzed. Interestingly, a captured metaphase FISH image showed that split 5′*CBFB* (red) and 3′*CBFB* (green) signals were located on different chromosomes, with evidence against inv(16) with *CBFB* rearrangement, as had been reported elsewhere. Mapback FISH was then performed with the same probe on a previously G-banded slide, and the results indicated the 5′*CBFB* (red) signal was located on a morphologically normal chromosome 16, the 3′*CBFB* (green) signal was located on an abnormal chromosome 2, and an intact *CBFB* signal (orange) was located on an abnormal chromosome 16 on its short (p) arm with material from chromosome 3 (Figure 3). RT-PCR testing performed on the BM showed no evidence of *CBFB::MYH11* transcripts. In the aggregate, these results show that *CBFB* rearrangement did not occur via *CBFB::MYH11* fusion; instead, *CBFB* had a novel partner most likely located on chromosome 2q. Sanger sequencing, in this case, also showed no evidence of *RAS* and *JAK2* mutation, and a PCR assay showed no evidence of FLT3 internal tandem duplication (ITD) or mutation.

### 3.3. Identification of CBFB::PPP1R7 Rearrangement by WGS and Sanger Sequencing

The low-pass whole genome sequencing (WGS) results confirmed the t(2;16), as detected by chromosomal and FISH analyses. Further analysis indicated the breakpoints were located between 241,662,723 to 241,662,798 of chromosome 2, falling within intron 7 of *PPP1R7*, and between 72,887,423 to 72,887,498 of chromosome 16, falling with the intron 5 of the *CBFB* gene.

To confirm the WGS results, PCR and Sanger sequencing were performed. Sanger sequencing of PCR products demonstrated *CBFB::PPP1R7* fusion. As shown in Figure 4, five selected clones submitted for Sanger sequencing demonstrated identical sequences. The 71 bases on the left side (541 to 611 bp) were blasted with a perfect match to the region of 241,662,758 to 241,662,828 of chromosome 2, while the 92 bases on the right side (609 to 700 bp) were blasted with a perfect match to the region of 72,887,284 to 72,887,377 of chromosome 16. Interestingly, the highlighted three bases (609 to 611) of CAG in the middle (Figure 4) were a microhomology (MH) shared by both chromosomes 2 and 16.

The breakpoints and the microhomology of CAG described were located in intron 5 of *CBFB* and intron 7 of *PPP1R7*, thus confirming what was postulated from the WGS results. The putative *CBFB*::*PPP1R7* fusion transcript was postulated to consist of exons 1 to 5 of *CBFB* and exons of 8 to 11 of *PPP1R7* (the variant 1 as example), whereas the reciprocal *PPP1R7::CBFB* transcript was postulated to be a recombination of exons 1 to 7 of *PPP1R7* and exon 6 of *CBFB*, as illustrated in Figure 5.

## 4. Discussion

In this study, we characterize *PPP1R7* as a novel partner of *CBFB* in a case of AML. In the literature, rare cases of *CBFB* rearrangement have been reported (Table 1). For example, Vendrame-Goloni et al. [21] reported a case with t(5;16)(q13;q22), and Gupta et al. [18] reported a case with t(1;16)(q43;q22). In the former case, FISH showed that the 5′*CBFB* signal remained on the affected chromosome 16, whereas the 3′*CBFB* signal was located on the abnormal chromosome 5. These cases are suggestive of the existence of new partners for *CBFB* rearrangement, although no CBFB-MYH11 RT-PCR was performed; thus, a potential cryptic *CBFB::MYH11* rearrangement, e.g., through insertion, followed by sequential t(5;16) or t(1;16) cannot be completely excluded. Nonetheless, there are very little data in the literature on novel partners of *CBFB* in cases of AML. Hence, we have reported this case in detail.

In addition to AML, *CBFB* rearrangements have been reported in patients with other types of malignant neoplasms involving the brain [28,29,30], breast [29], lung [29,30], oral cavity, gastrointestinal tract [29,30,31], skin [29], and ovary and uterus [29,30]. Several partner genes for *CBFB* have been identified in these malignancies, such as Abelson tyrosine-protein kinase 1 (*ABL1*) [28] and component of oligomeric golgi complex 4 (*COG4*) [29,30] in brain cancer; Tigger transposable element derived 7 (*TIGD7*) [29] in breast cancer; bromodomain-containing 9 (*BRD9*) [29,30] and nuclear transport factor 2 (*NUTF2*) [29] in lung cancer; ATPase 13A3 (*ATP13A3*) [29] in nasopharynx squamous cell carcinoma; *ATP13A3* [31] and NEDD8 activating enzyme E1 subunit 1 (*NAE1*) [29,30] in gastrointestinal tract cancer; carboxylesterase 4A (*CES4A*) [29,31], Gse1 coiled-coil protein (*GSE1*) [29], pleckstrin homology, Rho GEF domain-containing G4 (*PLEKHG4*) [29], carbonic anhydrase 7 (*CA7*) [29,30] in ovarian and uterine cancer; and ZFP90 zinc finger protein (*ZFP90*) [29] in skin cancer. As we searched the Mitelman Database of Chromosome Aberrations and Gene Fusions in Cancer (https://mitelmandatabase.isb-cgc.org; last accessed 20 June 2022) and PubMed (https://pubmed.ncbi.nlm.nih.gov/; last accessed 20 June 2022), *MYH11* was the only identified partner gene for *CBFB* rearrangement in AML. As reported previously by others and our group (Table 1), some AML cases were highly suspicious for a *CBFB* rearrangement, with novel partner gene(s), other than *MYH11*, especially those with t(1;16)(q21;q22), t(2;16), t(3;16), t(5;16), and t(16;19) with 16q22 involvement by conventional cytogenetic analyses; in some of these cases, CBFB FISH clearly demonstrated that the 3′*CBFB* signals were translocated to the partner chromosomes. However, to the best of our knowledge, *PPP1R7* identified in this study is the first novel partner gene of *CBFB* in AML since *MYH11* was identified in 1993 [5].

We retrospectively searched our database within a timeframe of 21 years (from 1 June 2000 through 31 May 2021) [24] and found only 10 AML cases showed clonal t(2;16) aberrations, but involving various band levels of both chromosome 2 and chromosome 16. Among the seven cases tested with CBFB BAP FISH, only the case included in this study showed a positive *CBFB* rearrangement. Therefore, the t(2;16) aberrations with/without *CBFB::PPP1R7* are extremely rare in AML. We also searched our database for the t(1;16)(q43;q22) [18], t(3;16)(q21;q22) [20], t(3;16)(q24;q22) [19], t(5;16)(q13;q22) [22], and t(5;16)(q33;q22) [22,23] that have been reported as rare *CBFB* rearrangement cases (Table 1). None of these cases has been found in our database. The clinicopathologic and cytogenetic features of all these cases with rare *CBFB* rearrangement, including our patient with t(2;16)/*CBFB::PPP1R7*, mimicked that of classic inv(16)/t(16;16) AML with *CBFB::MYH11* (Figure 1), except the AML case with t(1;16)(q43;q22)/*CBFB* rearrangement reported by Gupta et al. [18], which initially presented as a deceptive acute promyelocytic leukemia (APL) with multiple Auer rods in the abnormal promyelocytes, which has been finally diagnosed and treated as CBF AML. From a practice point of view, it should be pointed out that the t(2;16)(q37;q22) alteration in the case we report was subtle (Figure 2 and Figure 3). This abnormality could be easily missed when metaphase cells of low resolution are obtained for chromosomal analysis or misinterpreted as inv(16) when a positive CBFB BAP FISH result is obtained without a metaphase FISH image, as well as mapback FISH to show the exact locations of the 5′*CBFB* and 3′*CBFB* signals. A t(3;16)(p21;p13) was coexistent in our case. The low-pass WGS did not show any potential gene rearrangement(s) through this translocation. Therefore, the t(2;16)(q37;q22)/*CBFB::PPP1R7* rearrangement is likely the driver mutation in the AML case we report. The frequency of rare *CBFB* rearrangements is low in AML, but they can present in various forms of chromosomal aberrations with different partners. A confirmation of *CBFB* rearrangements by FISH test is necessary and relevant for clinical diagnosis and management in all these cases.

PPP1R7 protein, also called as Sds22, is an important unit of protein phosphatase-1 which physiologically regulates the activity and function of a serine/threonine phosphatase. PPP1R7 is also involved in signal transduction pathways, such as the activation of the cAMP-dependent protein kinase and GPCR pathways [32]. Chiang et al. also reported that *PPP1R7* knock-out caused cardiac dysfunction and disruption of calcium release from the sarcoplasmic reticulum in both animal models and cell lines [33]. PPP1R7 protein also plays an important role in accurate chromosome segregation [34,35,36,37], although the association between the PPP1R7 protein and carcinogenesis/tumorigenesis, especially leukemogenesis in hematologic malignancies, generally remains unknown. *PPP1R7* has been reported to be a tumor suppressor gene, since the allelic loss of 2q37, where this gene is seated, is associated with oral squamous cell carcinoma [38,39,40]. An in vitro study has shown that PPP1R7 can induce apoptosis and, consequently, inhibit breast cancer tumorigenesis, mainly through the negative regulation of the AKT signaling pathway [41]. In the current study, a microhomology of CAG at the break and reconnection sites of *CBFB* and *PPP1R7* was identified. As reported in the literature, microhomology-mediated end joining (MMEJ) is one of the mechanisms of double-strand DNA break repair. This process plays a physiological role in normal cells, but it is highly error-prone and can cause gene deletion and chromosomal rearrangement [42,43,44]. For example, microhomology-mediated recurrent deletions of *CALR, ASXL1,* and *SRSF2* and non-recurrent deletions in *TET2, DNMT3a, CEBPA*, and *RUNX1* have been reported in myeloid neoplasms [45]. In addition, microhomology-mediated recurrent *MYC* and other complex gene rearrangements have been reported in myeloma [46].

Segmental duplications (SDs), which are considered a common phenomenon and account for approximately 7% (or 218 Mbp) of the human genome [47], have been reported to be associated with the formation of chimeric fusion genes, such as *BCR::ABL1* through t(9;22)(q34;q11.2) [48] and *RUNX1::USP42* through ins(21;7)(q22;p15p22) [49], in myeloid malignancies. We searched the GenomeBrwoser (https://genome.ucsc.edu/. Last accessed 20 July 2022) but did not find any SDs in *PPP1R7*, *CBFB*, and their 5′ and 3′ flanking regions of approximately 20 kb. We analyzed our WGS data using the HMMcopy software, but a copy neutral status has been concluded in the same regions in our case. Although the karyotype analysis indicated a complex karyotype, including t(2;16) and t(3;16), and gain of chromosomes 8, 21, and 22, according to the morphologies of affected chromosome 2, 3, and 16, the FISH signal pattern and WGS analysis results, both the t(2;16) and t(3;16), were apparently balanced translocations, without microdeletions involving *PPP1R7*, *CBFB*, and their adjacent flanking regions. Therefore, the microhomology of CAG may explain the formation of t(2;16)(q37;q22)/*CBFB::PPP1R7* in this case.

One limitation of this study is that we are unable to characterize the t(2;16)(q37;q22)/*CBFB::PPP1R7* at the mRNA level, due to a lack of stocked purified RNA and suboptimal quality of RNA purified from paraffin-embedded tissue blocks. Therefore, we cannot explore the exact function(s) of *CBFB::PPP1R7* and how they may be applied to understand the pathogenesis of this case of AML.

In conclusion, we have successfully identified *PPP1R7* as a new partner of *CBFB* in a case of AML associated with t(2;16)(q37;q22). This case had clinicopathologic features of classic inv(16)/t(16;16) associated AML, suggesting that the novel *CBFB::PPP1R7* potentially plays a role similar to *CBFB*::*MYH11* in leukemogenesis.

## Figures and Tables

**Figure 1 genes-13-01367-f001:**
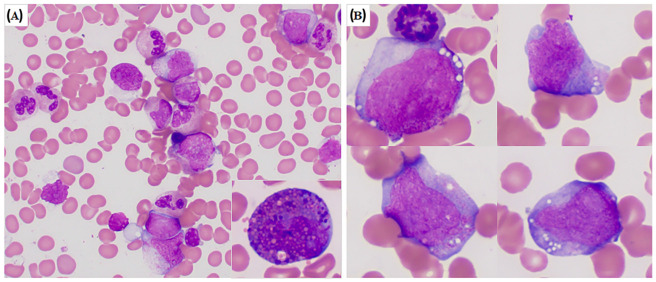
Bone marrow smear morphology of our patient with CBF AML through t(2;16)(q37;q22)/*CBFB::PPP1R7* rearrangement. (**A**). Representative area of bone marrow smears shows blasts, as well as maturing myeloid and monocytic cells. Rare abnormal eosinophils with basophilic granules were present (insert). (**B**). This composite figure illustrates the morphology of blasts. Most of the AML cells were large with round to irregular nuclei, dispersed chromatin, distinct nucleoli, and moderate amounts of basophilic cytoplasm. Some of the AML cells had cytoplasmic granules and vacuoles.

**Figure 2 genes-13-01367-f002:**
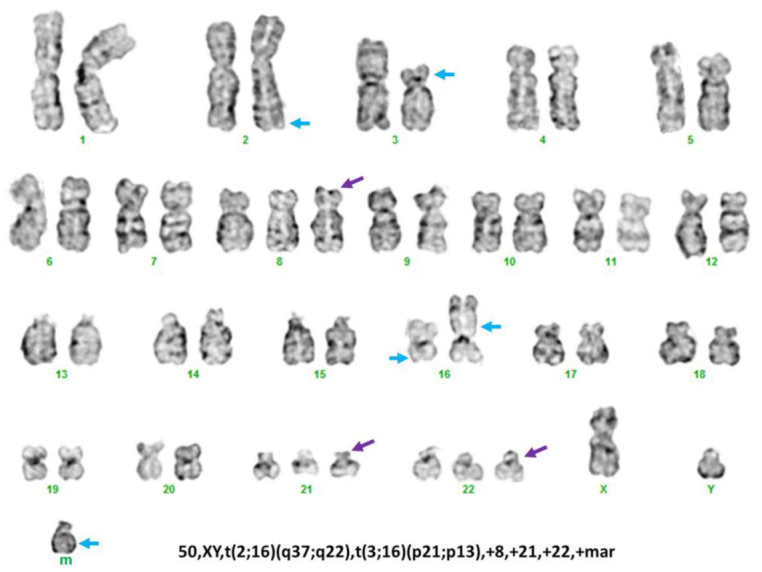
A representative karyogram of complex karyotype identified in our patient. The purple arrows indicate numerical abnormalities (+8, +21, +22) and the blue arrow indicates the structural aberrations involving 2q37, 3p21, 16q22, 16p13, and a marker chromosome.

**Figure 3 genes-13-01367-f003:**
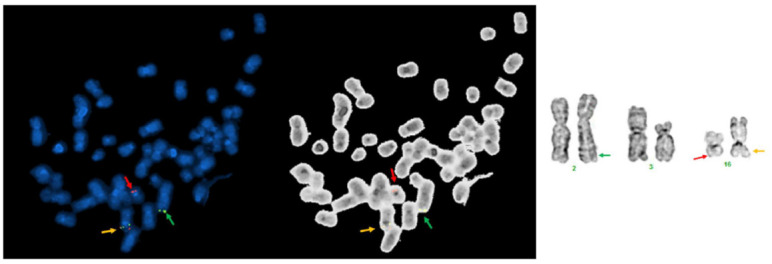
CBFB BAP FISH Images. A metaphase CBFB BAP FISH (**left**) and its inverted (**middle**) images are included. Each signal was also labelled on the corresponding chromosomes, as well as their band levels (**right**), to better understand the exact location of each signal. Red: 5′CBFB; green: 3′CBFB; orange: intact CBFB without rearrangement.

**Figure 4 genes-13-01367-f004:**
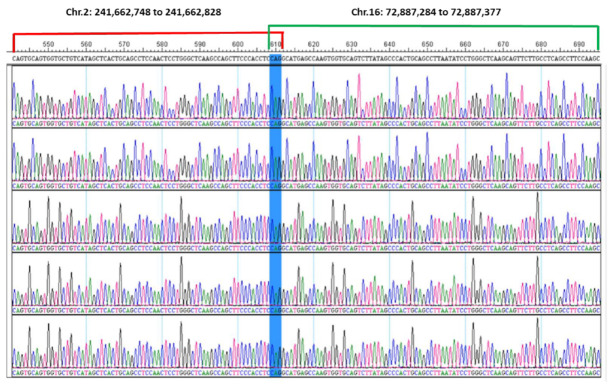
DNA Sanger sequencing result. Five selected clones submitted for Sanger sequencing were aligned here using the DNASTAR Lasergene 17 SeqManPro. They showed completely consensus sequences. The 71 bases on the left side (541 to 611 bp) were blasted to the region of 241,662,758 to 241,662,828 of chromosome 2, while the 92 bases on the right side (609 to 700 bp) were blasted to the region of 72,887,284 to 72,887,377 of chromosome 16. Both sides are with 100% of identities. The three bases of CAG (609 to 611) highlighted in blue were a microhomology shared by both chromosomes 2 and 16 in this case.

**Figure 5 genes-13-01367-f005:**
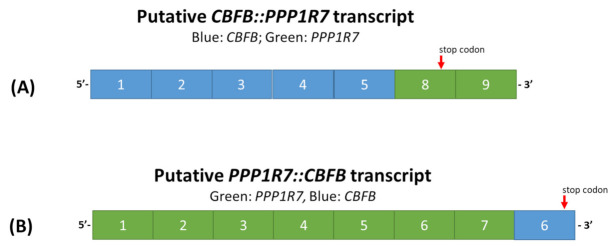
The putative *CBFB::PPP1R7* (**A**) and *PPP1R7::CBFB* (**B**) transcripts.

**Table 1 genes-13-01367-t001:** Case with potential novel partners for *CBFB* rearrangement reported in the literature.

Year	Reference #	Chr Aberrations	CBFB BAP FISH	Exclusion of *CBFB::MYH11*
1984	[19]	der(16)t(3;16)(q24;q22)	No FISH	no RT-PCR
1985	[20]	t(3;16)(q21;q22)	No FISH	no RT-PCR
1986	[22]	t(5;16)(q33;q22)	No FISH	no RT-PCR
1989	[23]	t(5;16)(q33;q22)	No FISH	no RT-PCR
1991	[16]	t(1;16) *	No FISH	no RT-PCR
2003	[21]	t(5;16)(q13;q22)	Yes. 3′CBFB + on 5q13	no RT-PCR
2021	[18]	t(1;16)(q43;q22)	Yes. 3′CBFB + on 1q43	no RT-PCR
2022	[24]	t(1;16)(q21;q22); t(2;16)(q37;q22); t(16;19)(q22;q13.3)	Yes. 3′CBFB + on 1q21, 2q37 and 19q13.3 respectively	Yes. RT-PCR negative

* Likely secondary event after an primary inv(16). Chr: chromosome; BAP: breakapart.

## Data Availability

Data presented in this study are available on request from the corresponding author.

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
