# Peer review of "PPP1R7* Is a Novel Translocation Partner of *CBFB* via t(2;16)(q37;q22) in Acute Myeloid Leukemia"

_genes, 2022, doi:10.3390/genes13081367_

Round 1
Reviewer 1 Report
The authors describe a rare case report of one patient with AML.
The AML genome is complex and definitely multiple cytogenetics and molecular techniques are required, because every technique has numerous limits. The authors correctly investigated the patient from the genetic point of view, however in order to increase the interest of the readers I recommend to describe in details the clinical aspects, evolution, prognostic etc .
Also I believe that more information from literature will be useful and the manuscript may be a structured as a review of the literature and a case report.
Author Response
Reply: we appreciated your review and comments. We reviewed and included all cases potentially with a novel partner for CBFB that have been published in the literature in the past decades (please see Table 1; Paragraph 1, Page 10). In the revised version, as you have suggested, we have emphasized the clinical aspects of rare CBFB rearrangements in AML and their clinical implication (please see paragraph 2, Page 11). As we indicated in our manuscript, previous similar studies didn’t characterize any novel partner gene at all. As you have suggested, we re-structured the revised manuscript, mainly in the discussion part.
Reviewer 2 Report
The Authors should investigate whether segmental duplications map near the genomic breakpoints of the genes involved in the rearrangement. In fact, as is known not only microhomologies but also segmental duplications are genomic elements able to mediate chromosomal rearrangements and cause genomic microdeletions in hematological malignancies (Albano F et al. Oncogene. 2010 29:2509-16; Zagaria A et al. Mol Cytogenet. 2014 7:66)
Author Response
Reply: Thank you for your input. You have a great point. We re-analyzed our WGS data for copy number status with emphasis in the regions of PPP1R7, CBFB and their adjacent flanking regions, and we didn’t find any microdeletion and/or microduplication in these regions in our case. We also search GenomeBrowser and didn’t find any SDs in the regions mentioned above. We think that the microhomology of CAG detected by sequencing may be the major cause of CBFB::PPP1R7 rearrangement in our case. Please see changes made in the revised manuscript by following your comments (Paragraph 1, Page 6; Paragraph 2, Page 12).
Reviewer 3 Report
This study was done under a previous study by the same group where they identified 3 cases of AML with t(1;16)(q21;q22), t(2;16)(q37;q22), or t(16;19)(q22;q13.3). The novelty of the current study is the identification of a novel partner PPP1R7 with CBFB.
1. This is a case study where the material and method section is well written.
2. The strategies implied to validate the chromosomal abnormality identified by karyotyping and FISH using WGS, Sanger sequencing, and RT-PCR are sufficient.
Author Response
Reply: Thank you very much for your review and comments.